# Determinants of healthcare insurance coverage among women of reproductive age in Tanzania: A multilevel mixed effect analysis

**Sanun Ally Kessy**[1][*], **Jovin R. Tibenderana**[1], **Jomo Gimonge**[2], **Fabiola V. Moshi**[2]

**1** Department of Epidemiology and Biostatistics, Kilimanjaro Christian Medical University College, Moshi, Tanzania, **2** Department of Clinical Nursing, The University of Dodoma, Dodoma, Tanzania

☯ These authors contributed equally to this work.

\* sahnunally@outlook.com

## Abstract

### Background

Health insurance has been one of the fundamental approaches of financing healthcare for sustainable Development goals (SDGs). There is a paucity of evidence on the determinants of healthcare insurance coverage among women of reproductive age in Tanzania, therefore this study aimed at assessing factors associated with access to health care insurance among women of reproductive age in Tanzania using national Demographic and health survey dataset.

### Methodology

This study was an analytical cross-sectional study that utilized nationally representative secondary data sourced from the Tanzania Demographic and Health Surveys (TDHS) of 2022. A weighted sample of 15,254 women of reproductive age (15–49) was included in the study. A multilevel regression analysis was used to examine factors associated with health care insurance coverage. These results were presented using adjusted odds ratio (AOR) with a 95% confidence interval.

### Results

In Tanzania the proportion of health insurance coverage among women of reproductive age was 5.8% whereby majority of women subscribing to employer's insurance (3.9%). In the fixed effect model, older women (35–49) were more likely to be covered with health insurance compared those aged 15–24 (AOR = 2.75, 95% CI: 2.19, 3.45). Similarly, married women had higher likelihood of health insurance coverage compared to unmarried (AOR = 1.60, 95% CI: 1.33, 1.92). Furthermore, educated women were more likely to be covered with health insurance than uneducated women (AOR = 6.96, 95% CI: 4.74, 10.22). Similarly, women who were exposed to mass media were more likely to be covered with health care insurance compared to unexposed (AOR = 1.43, 95% CI: 1.14, 1.78), in the Random effect model Intra-Cluster Correlation (ICC) value in Model III was 0.27

**Data Availability Statement:** The DHS data is freely available for download at https://dhsprogram.com/data/available-datasets.cfm.

**Funding:** The author(s) received no specific funding for this work.

**Competing interests:** The authors report no conflicts of interest in this work.

**Abbreviations:** AIC, Akaike Information Criteria; BIC, Bayesian Information Criteria CI, Confidence Interval; CHF, community health fund; DHS, Demographic and Health Survey; EA, Enumeration Area; HI, Health Insurance; LMIC, Low and Middle Income Country; NBS, National Bureau of Statistics: NSSF, National Social Security Fund; PR, Prevalence Risk; PSU, Primary Sampling Unit; SDGs, Sustainable development goals; UHC, Universal health coverage; VIF, Variance Inflation Factor; TIKA, Tiba Kwa Kadi; SHIB, Social Health Insurance Benefit; WRA, Women of Reproductive Age; TDHS, Tanzania Demographic and Health Survey.

## Conclusion

The low health insurance coverage among Tanzanian women of reproductive age reveals significant gaps in healthcare access Socioeconomic factors, along with limited access to reliable health information, highlight the importance of comprehensive and inclusive strategies to increase healthcare coverage. By implementing multifaceted approaches that tackle these root causes, Tanzania can improve the health and well-being of its women, ultimately contributing to a healthier and more resilient society.

## Background

Globally, health insurance has been one of the fundamental approaches of financing healthcare for sustainable Development goals (SDGs) by 2030. About 2 billion individuals are struggling with financial difficulties, with 1 billion encountering severe out of pocket [1]. Additionally, 344 million people globally are being pushed further into the burden of health care costs [1, 2]. Globally the significance of Universal Health Coverage (UHC) has been emphasized by its incorporation into the Sustainable Development Goals (SDGs) stating that, all individuals should have equal access to health care as right rather than a privilege with its target to be achieved by 2030 [3, 4].

Universal Health Coverage (UHC) means that all individuals and communities receive health services they need without suffering financial hardship. It includes the full spectrum of essential, quality health services, from health promotion to prevention, treatment, rehabilitation, and palliative care [2]. The global trajectory is deviating from the path to achieve substantial advancements in universal health coverage, a goal outlined in the Sustainable Development Goals (SDGs) target 3.8 set for 2030 which aim at protecting the vulnerable population from financial risks associated with severe health-related expenses [5].

UHC is important as it ensures provision of essential health services based on need, equitable access for the entire population, and the presence of financial-risk protection mechanisms to prevent economic hardship from healthcare expenses are key components of universal health coverage [6]. Women face considerable challenges in accessing medical services, primarily stemming from limited opportunities for economic empowerment and addressing their distinct healthcare requirements [5].

In Sub Saharan Countries, the prevalence of universal health coverage of health insurance coverage to be less than 10% [7, 8] The outcomes of Health Insurance (HI) implementation have been varying. Findings from research in Ghana [9]. and Rwanda [10] indicate that HI has contributed to greater equity in access to care for the overall population, alleviating the financial burden of healthcare expenses among the impoverished. Conversely, study findings from Ghana suggests Health Insurance (HI) did not provide uniform protection to all its members. Instead, it favored individuals with higher education, greater wealth, and those living in proximity to health facilities more than others [11].

Furthermore, among Sub-Saharan countries there is a notable correlation between formal sector employment and access to health insurance compared to informal employment sector [2]. Additionally, other factors such as such as income, education, household wealth status, marital status, age, and place of residence, as predictors of health insurance ownership [7, 12].

Despite notable accomplishments, the healthcare sector grapples with substantial hurdles in striving to deliver high-quality care to all individuals, regardless of their financial capacity to

afford services. Women encounter more pronounced obstacles in accessing critical healthcare services, including routine antenatal visits and childbirth services [5, 13, 14].

In Tanzania, over 80% of the population continues to rely on out-of-pocket payments for accessing healthcare services, often resulting in significant financial burdens [15]. Tanzania implemented different health insurance policies in the mid-1990s, such as National Health Insurance Fund (NHIF) in 1996 mandatory for every public servant and the Community Health Fund (CHF) in 2001the CHF, designed for the informal and rural communities, operates on a voluntary basis, offering coverage for up to six family members through a fixed annual payment, other health insurances such as Social Health Insurance Benefit (SHIB) established as a benefit under the National Social Security Fund (NSSF) and the Community Health Fund (CHF) and Tiba Kwa Kadi (TIKA) [15–17]

The capacity of women in the reproductive age group to attain affordable health coverage and care carries significant societal implications and hinges on the economic well-being of both women and their families, research have indicated that women often face a double burden of experiencing a higher prevalence of health issues while simultaneously being primarily responsible for caring for sick children and family members in African context. Nevertheless, the accessibility of healthcare services for women in the reproductive age group remains limited in Low- and Middle-Income Countries (LMIC), primarily due to financial burden [18].

In Tanzania, universal health coverage is in line with government policy of improving maternal health among reproductive age in order to improve maternal morbidities and mortality, more than 90% of men and women do not have any health insurance with only 5% coverage among children aged (0–14) years [19]. Despite several studies that have been done in Tanzania which were small scale, thus there is still a paucity of national wide information regarding health insurance utilizations. Additionally, there is a notable absence of published literature addressing health insurance coverage in the country. Hence to address this gap, the present study aimed to evaluate the health insurance coverage and its associated factors among women in the reproductive age group in Tanzania utilizing data from the Demographic and Health Survey (DHS). The findings of this study offer valuable insights for health planners, program developers, and healthcare professionals, providing a foundation for enhancing financial protection and service utilization among women of reproductive age.

## Methodology

### The data source

The study made use of the 2022 Tanzania Demographic and Health Survey (TDHS) data; a countrywide cross-sectional survey carried out every five years [20]. Tanzania, the largest country in East Africa, spans an area of 940,000 square kilometers, including 60,000 square kilometers of inland water. The study incorporated data from all regions of the United Republic of Tanzania as of 2022, the estimated population of Tanzania stood at 61,741,120, with an annual population growth rate of 3.2% [21, 22]. This study was an analytical cross-sectional study that utilized nationally representative secondary data sourced from the Tanzania Demographic and Health Surveys (TDHS) of 2022. The TDHS, is funded by the U.S Agency for International Development and implementation is carried out by the Ministry of Health (MoH) in Tanzania Mainland and Zanzibar, as well as the National Bureau of Statistics (NBS), the Office of the Chief Government Statistician (OCGS), with technical support provided by ICF International [21–24]. Measure Demographic and health survey (DHS) website was accessed to obtain the data (https://dhsprogram.com/)The data collection method of the surveys involves using a standardized questionnaire that is uniform across countries to gather information from women aged 15–49. The questionnaire is often translated into the primary

local languages of the participating nations. According to the DHS, these translated questionnaires, along with the original English version, undergo pretesting in both English and the local dialect to ensure their accuracy. Subsequently, during the pretest phase, field workers engage in an interactive discussion of the questions, providing suggestions for improvements across all versions. After the field practice, a debriefing session is conducted with the pretest field personnel, and adjustments are made to the questionnaires based on the insights gained from this process [20]. The survey employed face-to-face questionnaire interviews and utilized a stratified design with multistage cluster sampling to gather information on various aspects such as population health status, neonatal mortality, health behaviors, nutritional status, family planning, and demographics.

In the initial stage, 629 clusters were identified, from which households were then selected. Among these clusters, 26 households were systematically chosen to be representative from each cluster, resulting in a total of 16,354 households included in the survey. Eligibility for inclusion was based on the presence of all women aged 15–49 years in the selected household on the night prior to the interview. Further details on the sampling procedure and design have been previously documented [20]. A total of 15,254 women who had information on all the variables of interest were included in the study (Table 1). We adhered to the guidelines provided by the Strengthening the Reporting of Observational Studies in Epidemiology (STROBE) statement when writing the manuscript [25]. The DHS data is freely available for download at https://dhsprogram.com/data/available-datasets.cfm

**Study variables.** The outcome variable of this study was health insurance coverage. This was from the question "are you covered by any health insurance?". Response is coded as 0 = "No" and 1 = "Yes". The explanatory variables were age of the woman, marital status, education, employment, residence, wealth index, head of household sex, head of household age, media exposure, visited by healthcare worker for past12 months and visited health facility for the past 12 months. Age was recorded as 15–24, 25–34 and 35–49. Marital status was categorized as married and unmarried. Wealth status was categorized as poor, middle, and rich. Education was classified into three categories: no education, primary education, secondary education, or higher education. A new variable of media exposure was generated from household has either tv or radio. Our generated study variables and coding were based on previous literature [7, 26–29].

## Statistical analysis

We used Stata software, version 18.0 (Stata Corporation, College Station, TX, USA) for all statistical analyses. Initial presentation of health insurance coverage was done using percentages to show the proportion of women covered. Subsequently, through cross-tabulation, we explored the distribution of health insurance coverage among various background characteristics. Before conducting the regression analysis, we assessed potential collinearity among the variables using the variance inflation factor (VIF). Results indicated minimal collinearity, with VIF values ranging from 1.44 to 1.45 and a mean of 1.61, suggesting no significant multicollinearity concerns. To investigate the factors linked with health insurance coverage, we employed multilevel binary logistic analysis. This approach was chosen due to the complex survey design of the DHS, which involved a two-stage cluster sampling. Four models were utilized for analysis: Model 0: An empty model without any explanatory variables, Model I: Including individual-level factors, Model II: Incorporating community level factors and Model III: Considering all explanatory variables together. Crude and adjusted odds ratios (AOR) with corresponding 95% confidence intervals (CIs) were fit for each model. Both fixed and random effects were included in all models. The Intra-Cluster Correlation Coefficient (ICC) was used

**Table 1. Background characteristics and coverage of health insurance (Weighted).**

| Variables | | Covered with Health Insurance | |
|---|---|---|---|
| | Total (Percentage) | Yes (Percentage) | p-value |
| N | 15,254 (100.0%) | 888 (5.8%) | |
| **Age (years)** | | | |
| 15–24 | 5,810 (38.1%) | 263 (4.52%) | <0.001 |
| 25–34 | 4,609 (30.2%) | 273 (5.93%) | |
| 35–49 | 4,835 (31.7%) | 352 (7.28%) | |
| **Marital status** | | | |
| unmarried | 8,624 (56.5%) | 435 (5.05%) | <0.001 |
| married | 6,630 (43.5%) | 452 (6.82%) | |
| **Education** | | | |
| No education | 2,450 (16.1%) | 39 (1.57%) | <0.001 |
| Primary | 8,123 (53.3%) | 239 (2.94%) | |
| Sec/higher | 4,681 (30.7%) | 610 (13.04%) | |
| **Occupation** | | | |
| Not working | 5,452 (35.7%) | 277 (5.08%) | 0.032 |
| Working | 9,802 (64.3%) | 610 (6.23%) | |
| **Residence** | | | |
| Urban | 5,446 (35.7%) | 476 (8.75%) | <0.001 |
| Rural | 9,808 (64.3%) | 411 (4.19%) | |
| **Wealth index** | | | |
| Poor | 5,044 (33.1%) | 102 (2.01%) | <0.001 |
| Middle | 2,880 (18.9%) | 69 (2.39%) | |
| Rich | 7,330 (48.1%) | 717 (9.79%) | |
| **Sex of household head** | | | |
| Male | 10,918 (71.6%) | 595 (5.45%) | 0.017 |
| Female | 4,336 (28.4%) | 292 (6.74%) | |
| **Household head age** | | | |
| 15–24 | 539 (3.5%) | 20 (3.65%) | 0.089 |
| 25–34 | 3,245 (21.3%) | 172 (5.29%) | |
| 35–49 | 6,516 (42.7%) | 423 (6.49%) | |
| 50+ | 4,954 (32.5%) | 273 (5.50%) | |
| **Media Exposure** | | | |
| No | 6,653 (43.6%) | 191 (2.87%) | <0.001 |
| Yes | 8,601 (56.4%) | 696 (2.09%) | |
| **Visited by fieldworker in last 12 months** | | | |
| No | 14,781 (96.9%) | 844 (5.71%) | 0.017 |
| Yes | 473 (3.1%) | 43 (9.14%) | |
| **Visited health facility last 12 months** | | | |
| No | 7,167 (47.0%) | 329 (4.59%) | <0.001 |
| Yes | 8,087 (53.0%) | 558 (6.90%) | |

to quantify the proportion of variation in health insurance coverage attributable to differences between clusters (Primary Sampling Units, PSUs), measuring the similarity within clusters. Higher ICC values indicate greater within-cluster similarity, underscoring the need to account for clustering effects in the analysis. Fixed effects illustrated the relationship between explanatory variables and the outcome. Model fitness was evaluated using the Akaike Information Criterion (AIC), allowing us to determine how well each model fit the data, in adjusted analysis

we used stepwise regression to identify independent variables associated with health insurance coverage, A significance level of $p < 0.05$ was used for all statistical tests. Furthermore, we weighed all analyses to account for disproportionate sampling and non-response.

### Ethical clearance

Permission to download and utilize the data was obtained from the DHS Program/ICF International via http://www.dhsprogram.com. The data was strictly used for the current study's objectives. Extensive details regarding the methodology and ethical considerations of the DHS program can be found in the published reports of the Tanzania Demographic and Health survey [20, 24].

## Results

### Participants background characteristics

Among 15,254 participants of this study, the mean (SD) age was 29.3(9.8). The overall health insurance coverage among women of reproductive age was about 6%. Health insurance coverage was prevalent (7.28%) among women aged 35+ years. Around (6.82%) women who owned health insurance were married. The majority of women (13.0%) of study participants who were covered with health insurance had secondary or higher education. In occupation (6.23%) of participants who were insured were working. Less thana quarter (4.19%) of the study participants who owned health insurance were from rural areas (Table 1).

Most (9.79%) of women of reproductive age who owned health insurance were from rich household. In household (5.45%) of women of reproductive age who were insured were living in household whose head was male. Almost (6.49%) of the women who were insured were living in household whose head was aged 35–49 years of age. The majority (8.09%) of women who were insured had been exposed to media. Almost (9.14%) of the insured women had been visited by field worker in the last 12 months. Around (6.90%) of the study participants who are insured visited a health facility for the past 12 months (Table 1).

### Proportional of health insurance coverage

In Tanzania the proportion of health insurance coverage among women of reproductive age is 5.8%. The rest of the group are not covered with any health insurance (Fig 1)

### Specific types of health insurance coverage

Fig 2 shows percentage of women with specific types of health insurance. Most of them (5.8%) have any insurance followed by employers' insurance (3.9%) and the lowest (0.2%) being social security insurance (Fig 2).

### Determinants of health insurance coverage among Tanzanian reproductive age women

**Fixed effect.** Table 2, Model III presents determinants of health insurance coverage among Tanzanian reproductive age women. Older women (35–49) were more likely to be covered with health insurance compared to those aged 15–24 (AOR = 2.75, 95% CI: 2.19, 3.45). Similarly, married women had higher likelihood of health insurance coverage compared to unmarried (AOR = 1.60, 95% CI: 1.33, 1.92). The odds of health insurance coverage were higher among educated women (secondary/higher) compared to uneducated women (AOR = 6.96, 95% CI: 4.74, 10.22). Women who were exposed to media had higher odds of health insurance coverage compared to unexposed (AOR = 1.43, 95% CI: 1.14, 1.78). Women

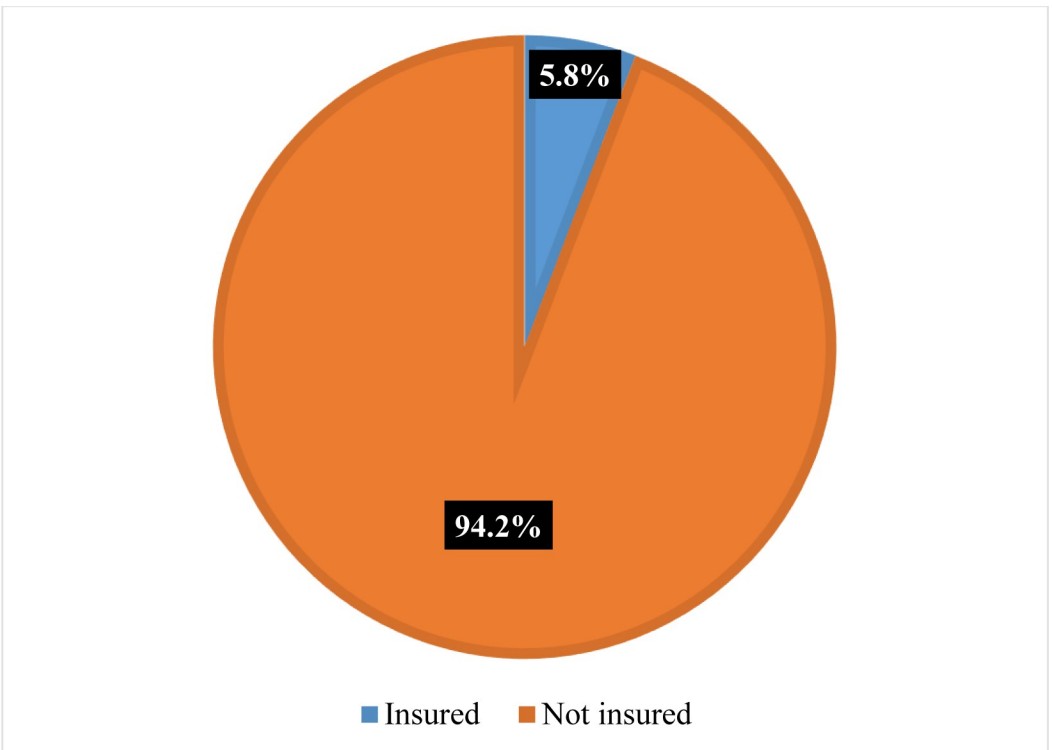

**Fig 1. Proportion of health insurance coverage among women of reproductive age.**

of reproductive age who visited health facility for the past 12 months had higher likelihood of health insurance coverage compared to those who didn't (AOR = 1.28, 95% CI: 1.09, 1.51). Additionally, participants from rich households were more likely to be covered with health

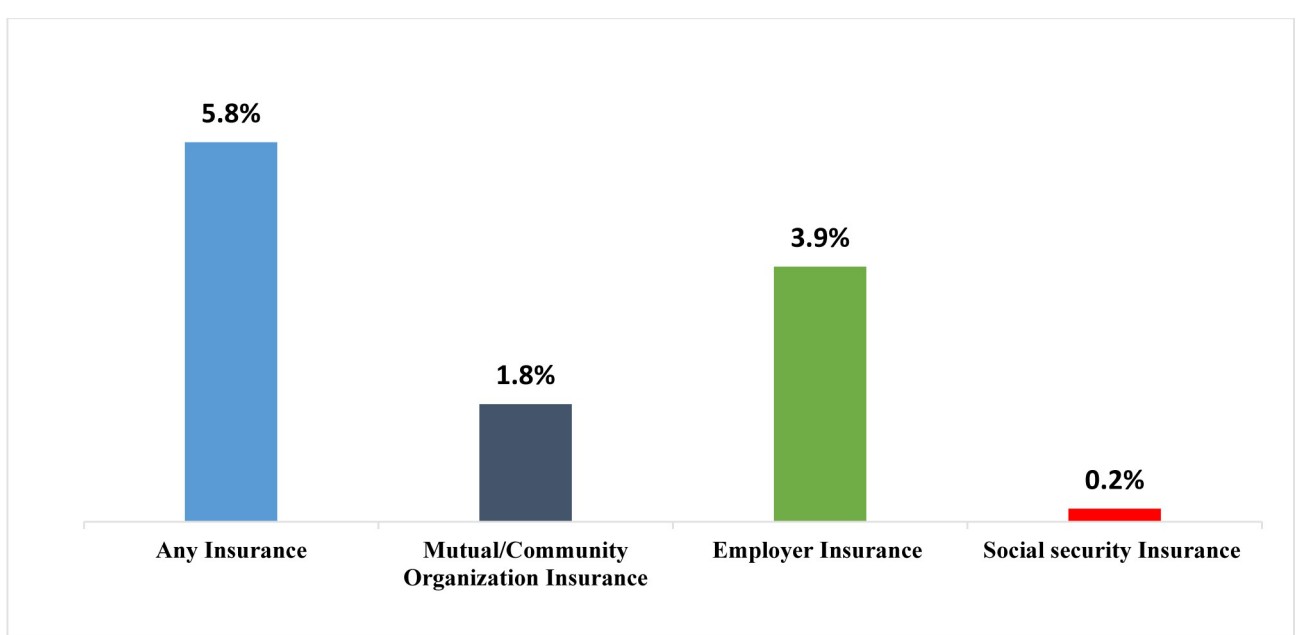

**Fig 2. Specific types of health insurance coverage.**

**Table 2. Multilevel analysis of determinants of health insurance coverage among reproductive age women in Tanzania (N = 15,254).**

| Variable | Model 0 | Model I AOR (95% CI) | Model II AOR (95% CI) | Model III AOR (95% CI) |
|---|---|---|---|---|
| **Age (years)** | | | | |
| 15–24 | | 1 | | 1 |
| 25–34 | | 1.41*(1.14, 1.74) | | 1.30* (1.05,1.61) |
| 35–49 | | 3.12*** (2.49, 3.90) | | 2.75*** (2.19,3.45) |
| **Marital status** | | | | |
| unmarried | | 1 | | 1 |
| married | | 1.41*** (1.19, 1.68) | | 1.60*** (1.33,1.92) |
| **Education** | | | | |
| No education | | 1 | | 1 |
| Primary | | 1.91**(1.32, 2.74) | | 1.57**(1.08,2.27) |
| Sec/higher | | 9.91*** (6.8, 14.39) | | 6.96*** (4.74,10.22) |
| **Occupation** | | | | |
| Not working | | 1 | | 1 |
| Working | | 1.02(0.86, 1.21) | | 0.98(0.82,1.17) |
| **Media Exposure** | | | | |
| No | | 1 | | 1 |
| Yes | | 2.02*** (1.67, 2.46) | | 1.43**(1.14,1.78) |
| **Visited by fieldworker in last 12 months** | | | | |
| No | | 1 | | 1 |
| Yes | | 1.29(0.89, 1.86) | | 1.27(0.88,1.84) |
| **Visited health facility last 12 months** | | | | |
| No | | 1 | | 1 |
| Yes | | 1.28*(1.09, 1.51) | | 1.28**(1.09,1.51) |
| **Residence** | | | | |
| Urban | | | 1 | 1 |
| Rural | | | 0.97(0.75, 1.25) | 1.00(0.76,1.33) |
| **Wealth index** | | | | |
| Poor | | | 1 | 1 |
| Middle | | | 1.20(0.86, 1.69) | 0.83(0.58,1.17) |
| Rich | | | 6.19*** (4.67, 8.21) | 2.78*** (1.99,3.87) |
| **Sex of household head** | | | | |
| Male | | | 1 | 1 |
| Female | | | 1.24*** (1.06, 1.45) | 1.42*** (1.19,1.70) |
| **Random effect** | | | | |
| ICC (%) | 29% | 26% | 24% | 27% |
| Wald Chi-square | | 524.55 | 260.83 | 573.75 |
| Mean VIF | | 1.45 | 1.44 | 1.61 |
| **Model fitness** | | | | |
| Log-likelihood | -3144.76 | -2837.71 | -2991.92 | -2786.80 |
| AIC | 6293.52 | 5697.43 | 5995.83 | 5603.60 |
| N | 15,254 | 15,254 | 15,254 | 15,254 |
| Number of clusters | 628 | 628 | 628 | 628 |

AOR = adjusted odds ratios; CI = Confidence Interval

*p < 0.05

**p < 0.01

***p < 0.001; 1 = Reference category; ICC = Intra-Class Correlation coefficient; AIC = Akaike Information Criterion; N = Total sample size

insurance compared to their counterparts (AOR = 2.78, 95% CI: 1.99, 3.87). We found that women who are living in households lead by female.

**Random effect.**   Table 2 presents results for random effect. We observed that the Intra-Cluster Correlation (ICC) value in Model III was 0.27 indicating that approximately 27% of the variability in health insurance coverage is attributed to differences between the clusters. The cluster-level variation, as indicated by the ICC values were 29%, 26%, 24% and 27% for Model 0, Model I, Model II and Model III respectively. We opted to focus on interpreting the results of Model III for discussion, as it had the lowest AIC value (5603.60) among the models, indicating a goodness of fit.

## Discussion

This study aimed to analyze the determinants affecting the possession of health insurance among Tanzanian women as a key approach to achieving universal health coverage. It found that insignificant majority of women of about 5.8% own any form of health insurance. The research identified several determinants associated with health insurance coverage, including age, economic status, educational attainment, marital status, and mass media exposure, Visited health facility last 12 months.

Our study revealed that only 5.8% of Tanzanian women possess health insurance, figure is contrary with Ghana having more than half percent accessing health healthcare [30]. These findings are in line with less than one tenth of the women of the reproductive age in Mauritania, Ethiopia and Nigeria [8, 26, 31]. This disparity can be explained that in Ghana there is a difference in health policy agendas, like livelihood empowerment against Poverty Program (LEAP) in Ghana [30] A pro-poor intervention policy, use of Participatory welfare ranking (PWR) and geographic targeting (GT) for poor groups helped in improving efficiency and equity needs under NHIS program in Ghana. It has been mandatory and compulsory for all Ghanaians to join health insurance scheme in Ghana [30].

This study found that compared to being unmarried, being married was associated with having health insurance coverage. Studies have identified that being in a dual income household offers more avenues for health insurance, similar findings have been reported in other studies conducted in East Africa, Zambia and Kenya [7, 29, 32] but inconsistence with the study done in Malawi [4]. Additionally, unmarried women may struggle to afford insurance premiums and deductibles and may perceive their risk of illness as lower [7].

The study suggests that women 35 years of age and older are more likely to subscribe to healthcare insurance, which is in line with findings from several other African nations, such as South Africa, and Nigeria and the one done in sub–Saharan Africa [2, 5, 33]. There are several reasons for this trend. Firstly, people in this age group typically find employment, which makes it possible for them to pay for health insurance. Furthermore, older women may have more children who are more likely to become sick and frequently take on caregiving responsibilities, which may lead them to look into insurance to reduce their out-of-pocket costs.

Women's education level had a substantial impact on whether or not they had health insurance. Individuals with more educational attainment were shown to be more likely than those without formal education to have health insurance. This finding adheres to other studies that show education is a significant factor in predicting one's ability to obtain health insurance [30, 34–36]. Education may help people obtain information more easily and increase their chances of entering high-paying professions, which are more likely to offer insurance coverage, which explains the positive association.

In comparison to women residing in the poor household, the current study showed that women residing in the rich household had greater odds of having health insurance coverage.

The findings of the national survey carried out in Ethiopia were consistent with these results [37], indicating a relationship between income levels and the probability of acquiring health insurance. In contrast, women from wealthier homes had lower probabilities of having health insurance coverage, according to different research conducted in Sudan [38].

The observed phenomenon of women with a history of visiting health facilities in the last 12 months exhibiting higher odds of having insurance coverage compared to their counterparts can be attributed to several factors. It is plausible that frequent visits to health facilities provide these women with increased exposure to healthcare services, thereby fostering a deeper understanding of the importance of insurance coverage and potentially reducing the barrier of out-of-pocket expenses. This finding aligns with existing research conducted in East African countries [7].

Furthermore, compared to women who were not exposed to the media, those who were exposed showed a higher likelihood to use health insurance. This finding is consistent with research from Ethiopia, Kenya, and Sub-Saharan Africa [30, 39, 40]. This can be explained by high coverage of internet and other media channel in the country which can act as an instrument for knowledge dissemination and raises awareness of the importance of health insurance.

## Strength and limitation

This study utilized the most current, nationally representative dataset, offering a comprehensive perspective on health insurance ownership among reproductive-age women. The demographic and health surveys (DHS) used are a valuable resource for addressing critical data gaps and supporting evidence-based health planning and interventions. Additionally, the large sample size ensures robust statistical power, enhancing the reliability and generalizability of the findings to the broader female population.

However, the study has notable limitations. As it relied on secondary data, the analysis was restricted to variables available in the DHS dataset, potentially excluding significant factors identified in other studies. The cross-sectional design further limits the ability to infer causality, as associations are observed at a single point in time without accounting for temporal dynamics. The reliance on self-reported data introduces potential biases, such as recall bias and social desirability bias, which may affect the accuracy of the findings. Furthermore, contextual or systemic factors, such as variations in regional health infrastructure or policies, could not be accounted for. Finally, unmeasured confounding variables, such as cultural attitudes toward insurance or the use of informal health services, may also influence the results but were beyond the scope of this study. Addressing these limitations in future research, such as through longitudinal designs or mixed-method approaches, could provide a more nuanced understanding of health insurance ownership among women.

## Conclusion

Less than ten percent of Tanzanian women of reproductive age (WRA) are registered in health insurance, representing an unacceptably low coverage rate and underscoring the urgent need to eliminate barriers to healthcare access for this population. Key determinants of health insurance coverage include marital status, wealth, education level, media exposure, and recent visits to medical facilities. These findings highlight the complex interplay between socioeconomic and informational factors in healthcare decision-making among WRA in Tanzania. Multifaceted strategies are required to address these challenges and increase health insurance coverage, contributing to the broader goal of achieving universal health coverage.

## Recommendations

To address the low health insurance coverage among WRA, promoting educational attainment, particularly in health and financial literacy, through comprehensive education initiatives and vocational training opportunities, is essential. Empowering widows and single women should also be a priority by ensuring they have access to resources and autonomy in making healthcare decisions. Leveraging mass media platforms to disseminate information about the benefits of health insurance can raise awareness and encourage adoption. Interventions must focus on addressing the root causes of low coverage to ensure equitable access to healthcare services across all population segments. Additionally, researchers should explore health insurance dropout rates through qualitative and quantitative studies to better understand the challenges and identify strategies for improving retention. These steps are crucial for policymakers, stakeholders, and researchers as Tanzania works toward achieving universal health coverage.

## Acknowledgments

The authors are grateful to MEASURE DHS for providing them with the data set.

## Author Contributions

**Conceptualization:** Sanun Ally Kessy, Jovin R. Tibenderana.

**Data curation:** Sanun Ally Kessy, Jovin R. Tibenderana.

**Formal analysis:** Sanun Ally Kessy, Jovin R. Tibenderana.

**Methodology:** Sanun Ally Kessy, Jovin R. Tibenderana.

**Supervision:** Fabiola V. Moshi.

**Visualization:** Sanun Ally Kessy, Jovin R. Tibenderana, Jomo Gimonge.

**Writing – original draft:** Sanun Ally Kessy, Jovin R. Tibenderana, Jomo Gimonge.

**Writing – review & editing:** Sanun Ally Kessy, Jovin R. Tibenderana, Jomo Gimonge, Fabiola V. Moshi.

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
