## [Decision Letter · Decision Letter 0]

5 Nov 2024

PONE-D-24-14959Determinants of Healthcare Insurance Coverage among Women of Reproductive Age in Tanzania: A Multilevel Mixed Effect Analysis.PLOS ONE

Dear Dr. Kessy,

Thank you for submitting your manuscript to PLOS ONE. After careful consideration, we feel that it has merit but does not fully meet PLOS ONE’s publication criteria as it currently stands. Therefore, we invite you to submit a revised version of the manuscript that addresses the points raised during the review process. Please submit your revised manuscript by Dec 20 2024 11:59PM. If you will need more time than this to complete your revisions, please reply to this message or contact the journal office at plosone@plos.org. Please include the following items when submitting your revised manuscript:A rebuttal letter that responds to each point raised by the academic editor and reviewer(s). You should upload this letter as a separate file labeled 'Response to Reviewers'.A marked-up copy of your manuscript that highlights changes made to the original version. You should upload this as a separate file labeled 'Revised Manuscript with Track Changes'.An unmarked version of your revised paper without tracked changes. You should upload this as a separate file labeled 'Manuscript'.

We look forward to receiving your revised manuscript.

Kind regards,

Devaraj Acharya, Ph.D.

Academic Editor

PLOS ONE

Journal Requirements:

2. Please note that your Data Availability Statement is currently missing the repository name. If your manuscript is accepted for publication, you will be asked to provide these details on a very short timeline. We therefore suggest that you provide this information now, though we will not hold up the peer review process if you are unable.

Additional Editor Comments:

After carefully assessing your manuscript, I decided that it requires a ‘Major Revision.’ Here are some points, please consider these before resubmission:

1. Citation: Ensure that all citations are correctly formatted and adhere to the guidelines of journal for article format.

2. Policy Relevancy: Add health insurance related policy to your paper to enhance its significance.

3. Recent Studies: Consult and compare your findings with the most recent studies.

4. Reviewers’ Comments: Address all comments made by the reviewers.

Reviewers' comments:

Reviewer's Responses to Questions

**Comments to the Author**

1. Is the manuscript technically sound, and do the data support the conclusions?

Reviewer #1: Yes

Reviewer #2: No

Reviewer #3: Partly

Reviewer #4: Partly

2. Has the statistical analysis been performed appropriately and rigorously? 

Reviewer #1: Yes

Reviewer #2: N/A

Reviewer #3: No

Reviewer #4: No

3. Have the authors made all data underlying the findings in their manuscript fully available?

Reviewer #1: Yes

Reviewer #2: Yes

Reviewer #3: Yes

Reviewer #4: Yes

4. Is the manuscript presented in an intelligible fashion and written in standard English?

Reviewer #1: Yes

Reviewer #2: Yes

Reviewer #3: No

Reviewer #4: Yes

5. Review Comments to the Author

Reviewer #1: Thank you for submitting your valuable work ,please add some data about health insurance coverage for men and children in your country .

also what about religion of the studied women dose it affect health insurance coverage in tanzania.

Reviewer #2: Dear Authors,

The area/field of research is important, locally, nationally, and globally, and I appreciate the research of the author to publish the paper and share the outcome with the scientific community.

Here are contradictory situations to accept the papers:

Title: Determinants of Healthcare Insurance Coverage among Women of Reproductive Age

in Tanzania: A Multilevel Mixed Effect Analysis

Objective: This study aimed at assessing factors

associated with access to health care insurance among women of reproductive age in

Tanzania using a national demographic and health survey dataset.

Situation: The Community Health Fund (CHF), designed for the informal and rural communities, operates on a voluntary basis, offering coverage for up to six family members through a fixed annual payment; other health insurances, such as Social Health Insurance Benefit (SHIB) established as a benefit under the National

Social Security Fund (NSSF) and the Community Health Fund (CHF) and Tiba Kwa Kadi (TIKA) (6,7).

So, if there is family based/household-based health insurance, all family members must have access to and an equal right to use health insurance benefits and packages regardless of women of reproductive age. So, it does make sense to analyze the data from women reproductive age.

Suggestion: In family/household-based health insurance, access to the health service is provided to family members regardless of age, gender, social status, or religion status. It is more worthy to present the health care utilization for those marginalized groups. Furthermore, access to the drop-out of health insurance and associated factors is important.

Good Luck !

Reviewer #3: The manuscript presents valuable insights for the field of health insurance research. However, I have a few specific recommendations to enhance its clarity and rigor.

Abstract:

* Results Section: Please include all methods of analysis used in the study, in addition to the multilevel analysis.

* Conclusion Section: Strengthen the concluding statements and avoid including percentages.

Data Collection Procedure:

* The authors analyze DHS data; it would be helpful to provide a summary of the data collection process, outlining the specific steps taken.

Variable Categorization:

* Please explain the criteria and sources used to categorize the study variables comprehensively.

Result

* The analysis in Table 1 contains errors. While the percentage for the "Yes" category (e.g., 5.8%) appears accurate, the percentages for other categories are incorrect. For instance, age-specific percentages should reflect values such as 4.5% among 15–24 years, 5.9% among 25–34 years, and 7.3% among 35–49 years. Kindly reanalyze and update Table 1, revising interpretations accordingly in the results section.

Discussion and Conclusion:

* The discussion and conclusion sections should be revised based on the corrected analysis.

Reviewer #4: The study is a great study to identify the factors that affect health insurance among women of reproductive age in Tanzania. Using the nationally representative sample to fulfill the objective of the study is a plus. However, the provided manuscript may require some changes, edits and revision on the analysis and writing part to further improve the strength and legibility.

Background:

1. The citations within the text do not seem to match with the references provided. This seems to be throughout the manuscript. For example, in the third paragraph there is jump from reference number 5 to reference number 14 and 23, 25 in fourth paragraph. I would like to suggest using reference managing software for proper citation and referencing, throughout the manuscript.

Methods:

1. As the study is the analysis of the secondary information available from the DHS survey, the methodology should be presented as an analysis of secondary information, rather than detailing about the DHS study setting and period, design, data collection procedure and sampling procedure and size (As DHS survey and your study that investigates DHS data are different). The DHS details however can be provided in a single subheading under data source.

2. The Demographic and Health Survey is a nationally representative survey (N=15,254) with high efforts gone towards development of variables. Same understanding on the authors’ part is also evident from the mentioning of DHS using the pretested standardized questionnaire within the “data collection procedure” subheading within the manuscript. Why are the variables with original categories in the DHS not used and recategorized then?

3. As the current study also includes the same sample size (15,254), is it possible that, for example married and unmarried categories add to that total (15,254) since the DHS also includes other categories like separated, divorced, widowed, etc. within the Marital status. Also, the references 7-10 do not relate to the statements made.

4. Assessing potential multicollinearity using VIF is commendable. However, use of variables in the final model must also be reassessed through other criteria when using logistic regression.

Results:

1. Model I, II and III includes variables which does not show significance for the adjusted Odds Ratio. What criterion has been used to include such non-significant variable in the final model. Were they significant when comparing one to one? Kindly go through the below articles and provide necessary corrections or response.

a. Bendel RB, Afifi AA. Comparison of Stopping Rules in Forward “Stepwise” Regression. J Am Stat Assoc 1977;72: 46–53. 5.

b. Costanza MC, Afifi AA. Comparison of Stopping Rules in Forward Stepwise Discriminant Analysis. J Am Stat Assoc 1979;74:777–85. http://dx.doi.org/10.1080/01621459.1979. 10481030.

c. Bursac, Z., Gauss, C.H., Williams, D.K. et al. Purposeful selection of variables in logistic regression. Source Code Biol Med 3, 17 (2008). https://doi.org/10.1186/1751-0473-3-17

Discussion:

1. This section includes all the findings from the study however the statements made by the authors (besides comparison with findings from other study) may require support from other studies, books, etc. For example: (Line 254-256) “This suggests that having a partner or spouse is beneficial, potentially because of the financial support from being in a dual-income household, which offers more avenues for acquiring health insurance.” Can be rewritten as “Studies have identified that being in a dual income household offers more avenues for health insurance” with citation.

2. Discussion, strength and limitation section and manuscript throughout may require some language review and editorial assistance.

Conclusion:

1. Are the variables with significant association influencing variables, factors, determinants or else. Kindly revisit the definition and use the same terminology throughout.

2. There may be many ways to empower women economically of which microfinance and income-generating businesses may be some. However, since they were not variables assessed within the study, these cannot directly be quoted into the conclusion for this study. Similar for other findings as well.

6. PLOS authors have the option to publish the peer review history of their article (what does this mean?). If published, this will include your full peer review and any attached files.

Reviewer #1: **Yes: **Seham Abdallah Elazab

Reviewer #2: No

Reviewer #3: No

Reviewer #4: No

---

## [Author Response · Author response to Decision Letter 0]

9 Nov 2024

Editor Comments:

Comment 1: Ensure that all citations are correctly formatted and adhere to the guidelines of journal for article format.

Response 1: Thank you for your comment we have ensured that all citations are properly formatted and adhere journal guidelines

Comment 2:Add health insurance related policy to your paper to enhance its significance.

Response 2: Thank you for your comment, we agree with this comment and we have added the insurance related policy to our paper and we believe it has enhanced its significance. 

Comment 3:Consult and compare your findings with the most recent studies.

Response 3:Thank you for the comment we have noted this and reshuffled our our comparison studies 

Reviewer 1

Comment 1: Please add some data about health insurance coverage for men and children in your country, also what about religion of the studied women does it affect health insurance coverage in Tanzania

Response 1: We appreciate your comment, we have added some data on health insurance coverage for men and children in Tanzania, additionally it’s to be noted that religion variable is not captured in Tanzania demographic health survey and hence could not be included in the analysis. 

Reviewer 2 

Comment 1: In family/household-based health insurance, access to the health service is provided to family members regardless of age, gender, social status, or religion status. It is more worthy to present the health care utilization for those marginalized groups. Furthermore, access to the drop-out of health insurance and associated factors is important.

Response 2: Thank you for your suggestion, in the background section we have included facts that, “The capacity of women in the reproductive age group to attain affordable health coverage and care carries significant societal implications and hinges on the economic well-being of both women and their families. Nevertheless, the accessibility of healthcare services for women in the reproductive age group remains limited in Low- and Middle-Income Countries (LMIC), primarily due to financial”, researches have indicated that women often face a double burden of experiencing a higher prevalence of health issues while simultaneously being primarily responsible for caring for sick children and family members in African context, refer to (https://www.afro.who.int/sites/default/files/2017-06/report-of-the-commission-on-womens-health-in-the-african-region---full-who_acreport-comp%20(1).pdf )

Since our study based on secondary data analysis and TDHS does not capture the drop out of health insurance coverage, therefore we have added in the recommendation section for other researchers to explore more on it by both qualitative and quantitative studies. 

 Reviewer 3 

Abstract

Comment 1:Results Section: Please include all methods of analysis used in the study, in addition to the multilevel analysis.

Response 1: Thank you for your recommendation, we have added all methods used in the study including multilevel analysis. 

Comment 2: Conclusion Section: Strengthen the concluding statements and avoid including percentages.

Response 2: Thank you for your recommendation, we have strengthened the concluding statement and we have not included the percentages. 

 Data Collection Procedure:

Comment 3: The authors analyze DHS data; it would be helpful to provide a summary of the data collection process, outlining the specific steps taken.

Response 3: We appreciate your recommendation; we have addressed the concern in Line 133-155 explaining about the data collection process and specific steps taken. 

Comment 4: Please explain the criteria and sources used to categorize the study variables comprehensively

Response 4: Thank you for your concern, the criteria used to categorize the study variables was informed based on the previous literature as comprehensively explained in manuscript in Line 157-166.

Results 

Comment 5 & 6: The analysis in Table 1 contains errors. While the percentage for the "Yes" category (e.g., 5.8%) appears accurate, the percentages for other categories are incorrect. For instance, age-specific percentages should reflect values such as 4.5% among 15–24 years, 5.9% among 25–34 years, and 7.3% among 35–49 years. Kindly reanalyze and update Table 1, revising interpretations accordingly in the results section .Discussion and Conclusion: The discussion and conclusion sections should be revised based on the corrected analysis.

Response 5 & 6: Thank you for the observation, We have re-analyzed and updated table 1, lastly, we have revised discussion and conclusion sections based on the corrected analysis.

Reviewer 4

Background:

Comment 1: The citations within the text do not seem to match with the references provided. This seems to be throughout the manuscript. For example, in the third paragraph there is jump from reference number 5 to reference number 14 and 23, 25 in the fourth paragraph. I would like to suggest using reference managing software for proper citation and referencing, throughout the manuscript.

Response 1: Thank you for the constructive comment and observation, we have put the citations and they are currently matching and we have used reference manager for citation and referencing throughout the manuscript. 

Methods:

Comment 2:. As the study is the analysis of the secondary information available from the DHS survey, the methodology should be presented as an analysis of secondary information, rather than detailing about the DHS study setting and period, design, data collection procedure and sampling procedure and size (As DHS survey and your study that investigates DHS data are different). The DHS details however can be provided in a single subheading under data source

Response 2: Thank you for your comment, as suggested we reshuffle all the sub-titles in the method section and put them under one subtitle named “The data source”.

Comment 3: The Demographic and Health Survey is a nationally representative survey (N=15,254) with high efforts gone towards development of variables. Same understanding on the authors’ part is also evident from the mentioning of DHS using the pretested standardized questionnaire within the “data collection procedure” subheading within the manuscript. Why are the variables with original categories in the DHS not used and recategorized then?

Response 3: Thank you for your comment and critical observation, some of the variables were used as they were collected from DHS example residence,sex of health of household head among others but some were recategorized for better interpretation based on previous literatures which used DHS for data analysis for easier comparison in discussion section. 

Comment 4:As the current study also includes the same sample size (15,254), is it possible that, for example married and unmarried categories add to that total (15,254) since the DHS also includes other categories like separated, divorced, widowed, etc. within the Marital status. Also, the references 7-10 do not relate to the statements made.

Response 4: We appreciate your comment, marital status had two categories that are married/ in union and unmarried which combined divorced, never in union, no longer living together/separated, living with partner, widowed and hence adding up to the same total, and this is literature based categorization. Lastly we have updated the citation 7-10 and they now relate to the statement made. 

Comment 5: Assessing potential multicollinearity using VIF is commendable. However, use of variables in the final model must also be reassessed through other criteria when using logistic regression.

Response 5: Thank you for your comment, during the construction of the final model in our analysis we have employed the Akaike and Bayesian criterion information as stipulated in the line number 184-186. Importantly we used stepwise regression to include the variables in the final model, we have added the statement in our manuscript. 

Results:

Comment 6: Model I, II and III includes variables which do not show significance for the adjusted Odds Ratio. What criterion has been used to include such non-significant variables in the final model. Were they significant when comparing one to one? Kindly go through the below articles and provide necessary corrections or responses.

Response 6: Thank you for your comment, We used four(4) criterias to include the non-significant variables in the aforementioned models and these include 1. Literature( possible confounders) 2. Biological plausibility 3. Significance (p-value of less than 0.05) 4. Clinical relevance. 

Discussion:

Comment 7: This section includes all the findings from the study however the statements made by the authors (besides comparison with findings from other study) may require support from other studies, books, etc. For example: (Line 254-256) “This suggests that having a partner or spouse is beneficial, potentially because of the financial support from being in a dual-income household, which offers more avenues for acquiring health insurance.” Can be rewritten as “Studies have identified that being in a dual income household offers more avenues for health insurance” with citation.

Response 7: Thank you for your suggestions, we have rephrased the statement as suggested and we have added the citation to support it, thank you. 

Comment 8: Discussion, strength and limitation section and manuscript throughout may require some language review and editorial assistance.

Response 8:Thank you for your comment. We have done some language review to check for spelling and grammar and we have revised accordingly.

Conclusion:

Comment 9: Are the variables with significant association influencing variables, factors, determinants or else. Kindly revisit the definition and use the same terminology throughout.

Response 9: Thank you for your observation, we have revised accordingly and have used the term determinant throughout the manuscript. 

Comment 10: There may be many ways to empower women economically of which microfinance and income-generating businesses may be some. However, since they were not variables assessed within the study, these cannot directly be quoted into the conclusion for this study. Similar for other findings as well.

Response 10: Thank you for your comment, we have revised the manuscript and recommended it according to our analyzed findings. 

Thank you again for your valuable feedback. We believe that the revisions have significantly strengthened our manuscript, and we hope it will now meet the standards for publication in Plos One journal. 

Sincerely,

Sanun A Kessy

ORCID: https://orcid.org/0009-0007-6854-3887

---

## [Decision Letter · Decision Letter 1]

22 Nov 2024

PONE-D-24-14959R1Determinants of Healthcare Insurance Coverage among Women of Reproductive Age in Tanzania: A Multilevel Mixed Effect Analysis.PLOS ONE

Dear Dr. Kessy,

Thank you for submitting your manuscript to PLOS ONE. After careful consideration, we feel that it has merit but does not fully meet PLOS ONE’s publication criteria as it currently stands. Therefore, we invite you to submit a revised version of the manuscript that addresses the points raised during the review process.

We look forward to receiving your revised manuscript.

Kind regards,

Devaraj Acharya, Ph.D.

Academic Editor

PLOS ONE

Journal Requirements:

Reviewers' comments:

Reviewer's Responses to Questions

**Comments to the Author**

1. If the authors have adequately addressed your comments raised in a previous round of review and you feel that this manuscript is now acceptable for publication, you may indicate that here to bypass the “Comments to the Author” section, enter your conflict of interest statement in the “Confidential to Editor” section, and submit your "Accept" recommendation.

Reviewer #3: All comments have been addressed

Reviewer #4: (No Response)

2. Is the manuscript technically sound, and do the data support the conclusions?

Reviewer #3: Yes

Reviewer #4: Yes

3. Has the statistical analysis been performed appropriately and rigorously? 

Reviewer #3: Yes

Reviewer #4: Yes

4. Have the authors made all data underlying the findings in their manuscript fully available?

Reviewer #3: Yes

Reviewer #4: No

5. Is the manuscript presented in an intelligible fashion and written in standard English?

Reviewer #3: Yes

Reviewer #4: No

6. Review Comments to the Author

Reviewer #3: Thank you very much for resubmitting your paper for review. I appreciate the effort you have put into addressing my comments. I am pleased to note that most of the concerns I raised have been thoroughly addressed.

Reviewer #4: The manuscript reads better after addressing the comments from all the reviewers. However, it still has some areas for improvement and require some changes.

1. Revisit the editing aspect, throughout the manuscript. Missing “full stops” and other issues present throughout the text.

2. Still DHS methodology highlighted in the method section, which can be provided through a citation of the TDHS 2022. Methodology needs to highlight approach to the acquisition of the data, what variables were taken and why, how the variables were recategorized (marital status have more than two categories in the references stated)? The article cited within the manuscript (https://doi.org/10.1186/1475-9276-13-27) can be referred for the methodology section.

3. Also, for the ICC, can it be explained a bit more within the methodology. Have the cluster and strata been considered in the analysis, can it be elaborated?

4. Relating older women with children, education with high paying profession in the discussion may require some basis. For the media part, more than highlighting the country’s internet, focus should be on government and different stakeholders using the media to aware on the Health insurance and cite it accordingly.

5. Rewrite the strengths and limitation section. Sounds repetitive and unclear. There may be more limitations as well, one for example can be: unavailability of variables which were significant in other studies etc.

6. The conclusion still has non-related recommendations, that is, information on variables not analysed within the manuscript. Conclusion and recommendations need to be separated. Future studies on drop out from HI suggested in the conclusion which is irrelevant to this study.

7. Include DOIs or weblink in all the references.

7. PLOS authors have the option to publish the peer review history of their article (what does this mean?). If published, this will include your full peer review and any attached files.

Reviewer #3: **Yes: **Ramesh Adhikari

Reviewer #4: No

---

## [Author Response · Author response to Decision Letter 1]

28 Nov 2024

Dear editor, 

Thank you for the opportunity to revise and resubmit our manuscript titled “Determinants of Healthcare Insurance Coverage among Women of Reproductive Age in Tanzania: A Multilevel Mixed Effect Analysis.” (Manuscript ID: PONE-D-24-14959). We appreciate the constructive feedback provided by you and the reviewers, which has helped us improve our manuscript. Below, we address each of the comments raised. 

Reviewer 4

Comment 1: Revisit the editing aspect, throughout the manuscript. Missing “full stops” and other issues present throughout the text.

Response 1: Thank you for your feedback regarding the editing and formatting of the manuscript. we have thoroughly revisited the entire document to address all formatting issues, including missing full stops and other punctuation errors

Comment 2: Still DHS methodology highlighted in the method section, which can be provided through a citation of the TDHS 2022. Methodology needs to highlight approach to the acquisition of the data, what variables were taken and why, how the variables were recategorized (marital status have more than two categories in the references stated)? The article cited within the manuscript (https://doi.org/10.1186/1475-9276-13-27) can be referred for the methodology section.

Response 2: Thank you for your detailed feedback regarding the methodology section.

We have addressed the acquisition of data in the "Data Source" section, starting from line 112, where we have explicitly mentioned the use of the TDHS 2022 dataset and provided the relevant citation. Regarding the selection of variables, we clarified that variables were chosen based on prior literature, as cited within the manuscript (line 156). For the categorization of marital status, we have followed the approach recommended in the literature, including the reference DOI:10.3389/frhs.2022.780550, which provides a detailed framework for recategorization as well as follow your recommendation paper for look up and citation.

Comment 3: Also, for the ICC, can it be explained a bit more within the methodology. Have the cluster and strata been considered in the analysis; can it be elaborated?

Response 3: Thank you for your feedback. To address the role of the Intra-Cluster Correlation Coefficient (ICC) and clarify the consideration of clustering and stratification in the analysis, we have provided additional details. The ICC was used to quantify the proportion of variation in health insurance coverage attributable to differences between clusters (Primary Sampling Units, PSUs), measuring the similarity within clusters. Higher ICC values indicate greater within-cluster similarity, underscoring the need to account for clustering effects in the analysis. Both clusters (PSUs) and strata were incorporated into the analysis, with PSUs included as random effects to capture hierarchical data structures and between-cluster variation. Stratification variables were used in survey weighting to ensure representativeness and mitigate biases arising from the sampling design. Additionally, all analyses were weighted to account for disproportionate sampling and non-response, ensuring population-level representativeness. These considerations ensured that the complex survey design and hierarchical nature of the data were appropriately addressed.

Comment 4: Relating older women with children, education with high paying profession in the discussion may require some basis. For the media part, more than highlighting the country’s internet, focus should be on government and different stakeholders using the media to aware on the health insurance and cite it accordingly.

Response 4: Thank you for the valuable feedback. In response, we have written the discussion to ensure it remains evidence-based and concise. Women aged 35 years and older are more likely to subscribe to health insurance due to their increased likelihood of stable employment and caregiving responsibilities, which drive the need for financial risk protection. This explanation aligns with findings from South Africa, Nigeria, and other sub-Saharan African countries. Regarding education, rather than focusing on high-paying professions, we highlight its role in improving individuals’ ability to access and understand health-related information, which facilitates health insurance uptake. Finally, instead of emphasizing internet access, we now focus on how governments and stakeholders can effectively use media platforms to raise awareness about the benefits of health insurance through targeted campaigns. These revisions ensure a concise and evidence-supported discussion.

Comment 5: Rewrite the strengths and limitation section. Sounds repetitive and unclear. There may be more limitations as well, one for example can be: unavailability of variables which were significant in other studies etc.

Response 5: We appreciate your recommendation; we have addressed the concern and rewrite the strengths and limitation part.

Comment 6: The conclusion still has non-related recommendations, that is, information on variables not analysed within the manuscript. Conclusion and recommendations need to be separated. Future studies on drop out from HI suggested in the conclusion which is irrelevant to this study.

Response 6: Thank you for the observation, we have re-write the conclusion and recommendations part, as well as separate the conclusion from recommendations as advised

Comment 7: Include DOIs or weblink in all the references.

Response 7: Thank you for the constructive comment and observation, we have put the citations into proper Plos one format, and they are currently matching, and we have used reference manager for citation and referencing throughout the manuscript.

---

## [Editor Report · Decision Letter 2]

2 Dec 2024

Determinants of Healthcare Insurance Coverage among Women of Reproductive Age in Tanzania: A Multilevel Mixed Effect Analysis.

PONE-D-24-14959R2

Dear Dr. Kessy,

We’re pleased to inform you that your manuscript has been judged scientifically suitable for publication and will be formally accepted for publication once it meets all outstanding technical requirements.

Kind regards,

Devaraj Acharya, Ph.D.

Tribhuvan University

Academic Editor

PLOS ONE
---

## [Editor Report · Acceptance letter]

8 Dec 2024

PONE-D-24-14959R2 

PLOS ONE

Dear Dr. Kessy, 

I'm pleased to inform you that your manuscript has been deemed suitable for publication in PLOS ONE. Congratulations! Your manuscript is now being handed over to our production team.

Kind regards, 

on behalf of

Dr. Devaraj Acharya 

Academic Editor

PLOS ONE